# Combining Self-Training and Hybrid Architecture for Semi-supervised Abdominal Organ Segmentation

Wentao Liu[1], Weijin Xu[1], Songlin Yan[1], Lemeng Wang[1], Haoyuan Li[1], and Huihua Yang[1,2(✉)]

[1] School of Artificial Intelligence, Beijing University of Posts and Telecommunications, Beijing 100876, China
yhh@bupt.edu.cn
[2] School of Computer Science and Information Security, Guilin University of Electronic Technology, Guilin 541004, China

**Abstract.** Abdominal organ segmentation has many important clinical applications, such as organ quantification, surgical planning, and disease diagnosis. However, manually annotating organs from CT scans is time-consuming and labor-intensive. Semi-supervised learning has shown the potential to alleviate this challenge by learning from a large set of unlabeled images and limited labeled samples. In this work, we follow the self-training strategy and employ a high-performance hybrid architecture (PHTrans) consisting of CNN and Swin Transformer for the teacher model to generate precise pseudo labels for unlabeled data. Afterward, we introduce them with labeled data together into a two-stage segmentation framework with lightweight PHTrans for training to improve the performance and generalization ability of the model while remaining efficient. Experiments on the validation set of FLARE2022 demonstrate that our method achieves excellent segmentation performance as well as fast and low-resource model inference. The average DSC and NSD are 0.8956 and 0.9316, respectively. Under our development environments, the average inference time is 18.62 s, the average maximum GPU memory is 1995.04 MB, and the area under the GPU memory-time curve and the average area under the CPU utilization-time curve are 23196.84 and 319.67. The code is available at https://github.com/lseventeen/FLARE22-TwoStagePHTrans.

**Keywords:** Abdominal Organ segmentation · Semi-supervised learning · Hybrid architecture.

## 1 Introduction

Medical image segmentation aims to extract and quantify regions of interest in biological tissue or organ images. Among them, abdominal organ segmentation has many important clinical applications, such as organ quantification, surgical planning, and disease diagnosis. However, manually annotating organs from CT

scans is time-consuming and labor-intensive. Thus, we usually cannot obtain a huge number of labeled cases. As a potential alternative, semi-supervised semantic segmentation has been proposed to learn a model from a handful of labeled images along with abundant unlabeled images to explore useful information from unlabeled cases. The organizer of FLARE2022 curated a large-scale and diverse abdomen CT dataset, including 2300 CT scans from 20+ medical groups. There are 50 labeled data and 2000 unlabeled data available. Compared with FLARE 2021, the challenge for FLARE 2022 is how to leverage the large amount of unlabeled data to improve the segmentation performance while taking into account efficient inference.

Self-training [8] via pseudo labeling is a conventional, simple, and popular pipeline to leverage unlabeled data, where the retrained student is supervised with hard labels produced by the teacher trained on labeled data, which is commonly regarded as a form of entropy minimization in semi-supervised learning [17]. The performance of the model of teacher and student in it is crucial. Benefiting from the excellent representation learning ability of deep learning, convolutional neural networks [13,1] (CNNs) have achieved tremendous success in medical image analysis. In spite of achieving extremely competitive results, CNN-based methods lack the ability to model long-range dependencies due to inherent inductive biases such as locality and translational equivariance. Transformer [16,3,10], relying purely on attention mechanisms to model global dependencies without any convolution operations, has emerged as an alternative architecture that has delivered better performance than CNNs in computer vision on the condition of being pre-trained on large-scale datasets. Therefore, many hybrid architectures derived from the combination of CNN and Transformer have emerged, which offer the advantages of both and have gradually become a compromise solution for medical image segmentation without being pre-trained on large datasets.

In previous work [9], we proposed a parallel hybrid architecture (PHTrans) for medical image segmentation where the main building blocks consist of CNN and Swin Transformer to simultaneously aggregate global and local representations. PHTrans can independently construct hierarchical local and global representations and fuse them in each stage, fully exploiting the potential of CNN and the Transformer. Extensive experiments on BCV [7] demonstrated the superiority of PHTrans against other competing methods on abdominal multi-organ segmentation. In this work, we propose a solution combining the hybrid architecture PHTrans with self-training for the FLARE2022 challenge. Firstly, we employ a high-performance PHTrans with the nnU-Net frame as the teacher model to generate precise pseudo-labels for unlabeled data. Secondly, the labeled data and pseudo-labeled data are fed together into a two-stage segmentation framework with lightweight PHTrans for training, which locates the regions of interest (ROIs) first and then finely segments them. Experiments on the validation set of FLARE2022 demonstrate that our method achieves excellent segmentation performance as well as fast and low-resource model inference.

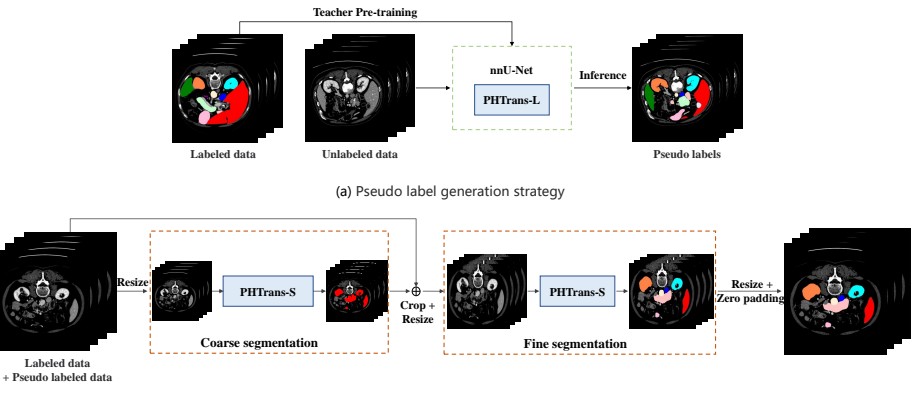

(a) Pseudo label generation strategy

(b) Two-stage segmentation frame

**Fig. 1.** Overview of our proposed semi-supervised abdominal organ segmentation method based on self-training and PHTrans.

## 2    Method

We propose a semi-supervised abdominal organ segmentation method based on self-training [19] and PHTrans [9], as shown in Figure 1, which is a three-stage two-network (teacher and student) pipeline, i.e., (1) train a teacher model using labeled data, (2) generate pseudo labels for unlabeled data and (3) train a separate student model using labeled data and pseudo-labeled data. We employ one high-performance PHTrans and two lightweight PHTrans as teacher and student models, respectively. (1) and (2) are performed in the nn-UNet [6] framework with the default configuration except that UNet is replaced by PH-Trans. Some of the cases in the validation set include other tissues or organs in addition to the abdomen, such as the neck, buttocks, and legs. For instance, the last case in the validation set has 1338 slices, but only about 250 of those slices belong to the abdomen. Motivated by the solutions of Flare2021's champion and third winner [18,15], we adopt a two-stage segmentation framework and whole-volume-based input strategy in the student to improve the computational efficiency. We introduce the labeled data and the pseudo-labeled data together into the two-stage segmentation framework for training. The coarse segmentation model aims to obtain the rough location of the target organ from the whole CT volume. The fine segmentation model achieves precise segmentation of abdominal organs based on cropped ROIs from the coarse segmentation result. Finally, the segmentation result is restored to the size of the original data by resampling and zero padding. The method is described in detail in the following subsections.

### 2.1    Preprocessing

The preprocessing strategy for labeled data and pseudo-labeled data in the two-stage segmentation framework is as follows:

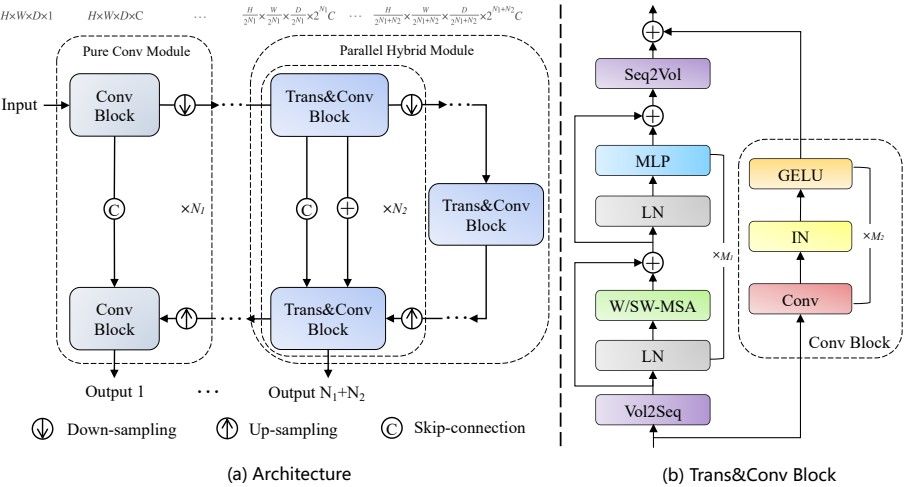

**Fig. 2.** (a) The architecture of PHTrans; (b) Parallel hybrid block consisting of Transformer and convolution (Trans&Conv block).

- Image reorientation to the target direction
- Resampling images to uniform sizes. We use small-scale images as the input of the two-stage segmentation to improve the segmentation efficiency. Coarse input: [64, 64, 64]; Fine input: [96, 192, 192].
- We applied a z-score normalization based on the mean and standard deviation of the intensity values in the input volume.
- Considering that the purpose of coarse segmentation is to roughly extract the locations of abdominal organs, we set the voxels whose intensity values are greater than 1 in the resampled ground truth to 1, which converts the multiclassification abdominal organ segmentation into a simple two-classification integrated abdominal organ segmentation.

### 2.2   Proposed Method

An overview of the PHTrans [9] is illustrated in Figure 2 (a). PHTrans follows the U-shaped encoder and decoder design, which is mainly composed of pure convolution modules and parallel hybrid ones. Given an input volume $x \in \mathbb{R}^{H \times W \times D}$, where $H$, $W$ and $D$ denote the height, width, and depth, respectively, we first utilize several pure convolution modules to obtain feature maps $f \in \mathbb{R}^{\frac{H}{2^{N_1}} \times \frac{W}{2^{N_1}} \times \frac{D}{2^{N_1}} \times 2^{N_1}C}$, where $N_1$ and $C$ denote the number of modules and base channels, respectively. Afterwards, parallel hybrid modules consisting of Transformer and CNN were applied to model the hierarchical representation from the local and global feature. The procedure is repeated $N_2$ times with $\frac{H}{2^{N_1+N_2}} \times \frac{W}{2^{N_1+N_2}} \times \frac{D}{2^{N_1+N_2}}$ as the output resolutions and $2^{N_1+N_2}C$ as the channel number. Corresponding to the encoder, the symmetric decoder is similarly built based on pure convolution modules and parallel hybrid modules, and fuses

semantic information from the encoder by skip-connection and addition operations. Furthermore, we use deep supervision at each stage of the decoder during the training, resulting in a total of $N_1 + N_2$ outputs, where joint loss consisting of cross entropy and dice loss is applied. The architecture of PHTrans is straightforward and changeable, where the number of each module can be adjusted according to medical image segmentation tasks, i.e., $N_1$, $N_2$, $M_1$ and $M_2$. Among them, $M_1$ and $M_2$ are the numbers of Swin Transformer blocks and convolution blocks in the parallel hybrid module.

The parallel hybrid modules are deployed in the deep stages of PHTrans, where the Trans&Conv block, as its heart, achieves hierarchical aggregation of local and global representations by CNN and Swin Transformer. The scale-reduced feature maps are fed into Swin Transformer (ST) blocks and convolution (Conv) blocks, respectively. We introduce Volume-to-Sequence (V2S) and Sequence-to-Volume (S2V) operations at the beginning and end of ST blocks, respectively, to implement the transform of volume and sequence, making it concordant with the dimensional space of the output that Conv blocks produce. Specifically, V2S is used to reshape the entire volume (3D image) into a sequence of 3D patches with a window size. S2V is the opposite operation. As shown in Figure 2 (b), an ST block consists of a shifted window based multi-head self attention (MSA) module, followed by a 2-layer MLP with a GELU activation function in between. A LayerNorm (LN) layer is applied before each MSA module and each MLP, and a residual connection is applied after each module [10]. In $M_1$ successive ST blocks, the MSA with regular and shifted window configurations, i.e., W-MSA and SW-MSA, is alternately embedded into ST blocks to achieve cross-window connections while maintaining the efficient computation of non-overlapping windows.

For medical image segmentation, we modified the standard ST block into a 3D version, which computes self-attention within local 3D windows that are arranged to evenly partition the volume in a non-overlapping manner. Supposing $x \in \mathbb{R}^{H \times W \times S \times C}$ is the input of ST block, it would be first reshaped to $N \times L \times C$, where $N$ and $L = W_h \times W_w \times W_s$ denote the number and dimensionality of 3D windows, respectively. The convolution blocks are repeated $M_2$ times with a $3 \times 3 \times 3$ convolutional layer, a GELU nonlinearity, and an instance normalization layer (IN) as a unit. Finally, we fuse the outputs of the ST blocks and Conv blocks by an addition operation. The computational procedure of the Trans&Conv block in the encoder can be summarized as follows:

$$y_i = S2V(ST^{M_1}(V2S(x_{i-1}))) + Conv^{M_2}(x_{i-1}), \tag{1}$$

where $x_{i-1}$ is the down-sampling results of the encoder's $i-1^{th}$ stage. In the decoder, besides skip-connection, we supplement the context information from the encoder with an addition operation. Therefore, the Trans&Conv block in the decoder can be formulated as:

$$z_i = S2V(ST^{M_1}(V2S(x_{i+1} + y_i)) + Conv^{M_2}([x_{i+1}, y_i]), \tag{2}$$

where $x_{i+1}$ is the up-sampling results of the decoder's $i+1^{th}$ stage and $y_i$ is output of the encoder's $i^{th}$ stage. The down-sampling contains a strided convo-

lution operation and an instance normalization layer, where the channel number is halved and the spatial size is doubled. Similarly, the up-sampling is a strided deconvolution layer followed by an instance normalization layer, which doubles the number of feature map channels and halved the spatial size.

### 2.3   Post-processing

Connected component-based post-processing is commonly used in medical image segmentation. Especially in organ image segmentation, it often helps to eliminate the detection of spurious false positives by removing all but the largest connected component. We applied it to the output of the coarse and fine models.

## 3    Experiments

### 3.1   Dataset and evaluation measures

The FLARE 2022 is an extension of the FLARE 2021 [11] with more segmentation targets and more diverse abdomen CT scans. The FLARE2022 dataset is curated from more than 20 medical groups under the license permission, including MSD [14], KiTS [4,5], AbdomenCT-1K [12], and TCIA [2]. The training set includes 50 labeled CT scans with pancreas disease and 2000 unlabeled CT scans with liver, kidney, spleen, or pancreas diseases. The validation set includes 50 CT scans with liver, kidney, spleen, or pancreas diseases. The testing set includes 200 CT scans where 100 cases has liver, kidney, spleen, or pancreas diseases and the other 100 cases has uterine corpus endometrial, urothelial bladder, stomach, sarcomas, or ovarian diseases. All the CT scans only have image information and the center information is not available.

The evaluation measures consist of two accuracy measures: Dice Similarity Coefficient (DSC) and Normalized Surface Dice (NSD), and three running efficiency measures: running time, area under GPU memory-time curve, and area under CPU utilization-time curve. All measures will be used to compute the ranking. Moreover, the GPU memory consumption has a 2 GB tolerance.

### 3.2   Implementation details

We employ two different configurations of PHTrans for pseudo-label generation and two-stage segmentation, respectively. In order to achieve high-precision pseudo-label generation, PHTrans-L adopted a high-performance configuration with large model parameters and computational complexity. In PHTrans-L, we empirically set the hyper-parameters $[N_1,N_2,M_1,M_2]$ to [2,2,2,2] and adopted the stride strategy of nnU-Net [6] for down-sampling and up-sampling. Moreover, the base number of channels C is 36, and the numbers of heads of multi-head self-attention used in different encoder stages are [3,6,12,24]. We set the size of 3D windows $[W_h,W_w,W_s]$ to [4,5,5] in ST blocks. However, PHTrans-S is configured as a lightweight architecture to meet efficient model inference, where the base

**Table 1.** The parameter setting of PHTrans-L and PHTrans-S.

| Model | PHTrans-L | PHTrans-S |
|---|---|---|
| Hyper-parameters $[N_1, N_2, M_1, M_2]$ | [2,4,2,2] | [2,3,2,2] |
| Base channel number | 36 | 16 |
| Down-sampling number | 5 | 4 |
| Heads number of self-attention | [3,4,12,24] | [4, 4, 4] |
| 3D windows size | [4,5,5] | [4, 4, 4]/[3, 4, 4] |
| MLP-ratio | 4 | 1 |

**Table 2.** Development environments and requirements.

| | |
|---|---|
| Windows/Ubuntu version | Ubuntu 20.04.3 LTS |
| CPU | Intel(R) Xeon(R) Silver 4214R CPU @ 2.40GHz |
| RAM | 32×4 GB; 2933 MT/s |
| GPU (number and type) | One NVIDIA 3090 24G |
| CUDA version | 11.6 |
| Programming language | Python 3.8.13 |
| Deep learning framework | Pytorch (Torch 1.11, torchvision 0.12.0) |
| Specific dependencies | nn-UNet |

number of channels is 16 and the number of up-sampling and down-sampling is 4. Other model hyperparameter settings are detailed in Table 1. The development environments and requirements are presented in Table 2. The training protocols for coarse and fine segmentation are presented in Table 3. In the training phase, we train on all 50 labeled data and random 450 pseudo-labeled data at each epoch. To alleviate the over-fitting of limited training data, we employed online data argumentation, including random rotation, scaling, adding white Gaussian noise, Gaussian blurring, adjusting rightness and contrast, simulation of low resolution, Gamma transformation, and elastic deformation.

## 4    Results and discussion

### 4.1    Quantitative results on validation set

Using the default nnU-Net as the baseline, we employ PHTrans instead of U-Net for comparative experiments. Table 4 shows that the average DSC on the validation set of "nnU-Net+PHTrans" is 0.8756, which is 0.0237 higher than that of nnU-Net, fully demonstrating PHTrans's excellent medical image segmentation ability. Although "nnU-Net+PHTrans" has achieved impressive segmentation results without using unlabeled data, nnU-Net is not conducive to the "Fast and Low-resource" of the FLARE challenge due to low-spacing resampling and sliding window inference. However, the segmentation results of "Two-stage+PHTrans" (TP) were less than satisfactory, with an average DSC of only 0.6889. We consider the reason is that the whole-volume based input strategy greatly reduces

**Table 3.** The training protocols of two-stage segmentation model.

| Model | Coarse model / Fine model |
|---|---|
| Network initialization | "he" normal initialization |
| Batch size | 64 / 4 |
| Patch size | 64×64×64 / 96×192×192 |
| Total epochs | 300 |
| Optimizer | AdamW |
| Initial learning rate (lr) | 0.01 |
| Lr decay schedule | Cosine Annealing LR |
| Loss function | Cross entropy + Dice |
| Training time (hours) | 0.5 / 19.75 |
| Number of model parameters | 6.66 M[3] |
| Number of flops | 18.60 / 251.19 G[4] |
| $CO_2$eq | 0.0856 / 1.7688 kg[5] |

**Table 4.** Ablation study on validation set. (PD: pseudo-labeled data participated in the training.)

| Methods | Mean DSC | Liver | RK | Spleen | Pancreas | Aorta | IVC |
|---|---|---|---|---|---|---|---|
| nnU-Net | 0.8519 | 0.9693 | 0.8662 | 0.9107 | 0.8433 | 0.9604 | 0.8813 |
| nnU-Net+PHTrans | 0.8756 | 0.9693 | 0.8997 | 0.9488 | 0.8601 | 0.9583 | 0.9161 |
| Two-stage+PHTrans | 0.6889 | 0.8930 | 0.7840 | 0.8307 | 0.5886 | 0.8828 | 0.8111 |
| Two-stage+PHTrans+PD | **0.8956** | **0.9761** | **0.9368** | **0.961** | **0.8728** | **0.9613** | **0.9165** |

| Methods | RAG | LAG | Gallbladder | Esophagus | Stomach | Duodenum | LK |
|---|---|---|---|---|---|---|---|
| nnU-Net | 0.8268 | 0.7806 | 0.7022 | 0.8558 | 0.8743 | 0.7384 | 0.8653 |
| nnU-Net+PHTrans | **0.8496** | 0.8178 | 0.7313 | **0.8777** | 0.9073 | 0.7822 | 0.8649 |
| Two-stage+PHTrans | 0.5774 | 0.4229 | 0.5535 | 0.6651 | 0.6782 | 0.5053 | 0.7632 |
| Two-stage+PHTrans+PD | 0.8132 | **0.835** | **0.8346** | 0.8737 | **0.9194** | **0.8045** | **0.9377** |

the resolution of the image and thus the segmentation accuracy is lost. In addition, the resampling volume of fixed size has a large difference in spacing, and it is difficult for the model to learn general representations in the limited labeled data, resulting in poor generalization ability. In contrast, we introduce pseudo-labeled data generated by the "nnU-Net+PHTrans" teacher model to further improve the segmentation performance and generalization ability. The results of "Two-stage+PHTrans+PD" (TPP) demonstrate that the original segmentation framework produces significant performance gains after training on 2000 pseudo-labeled data (PD). In spite of the fact that there is wrong label information in the pseudo labels, the average DSC improved from 0.6889 to 0.8956. Furthermore, table 5 shows that the TPP achieved an average NSD of 0.9316 on the validation set.

**Table 5.** Quantitative results of NSD on validation set. (PD: pseudo-labeled data participated in the training.)

| Methods | Mean NSD | Liver | RK | Spleen | Pancreas | Aorta | IVC |
|---|---|---|---|---|---|---|---|
| Two-stage+PHTrans+PD | 0.9316 | 0.9761 | 0.9198 | 0.9572 | 0.9422 | 0.9863 | 0.9138 |

| Methods | RAG | LAG | Gallbladder | Esophagus | Stomach | Duodenum | LK |
|---|---|---|---|---|---|---|---|
| Two-stage+PHTrans+PD | 0.9273 | 0.9125 | 0.8659 | 0.9258 | 0.9485 | 0.9158 | 0.9192 |

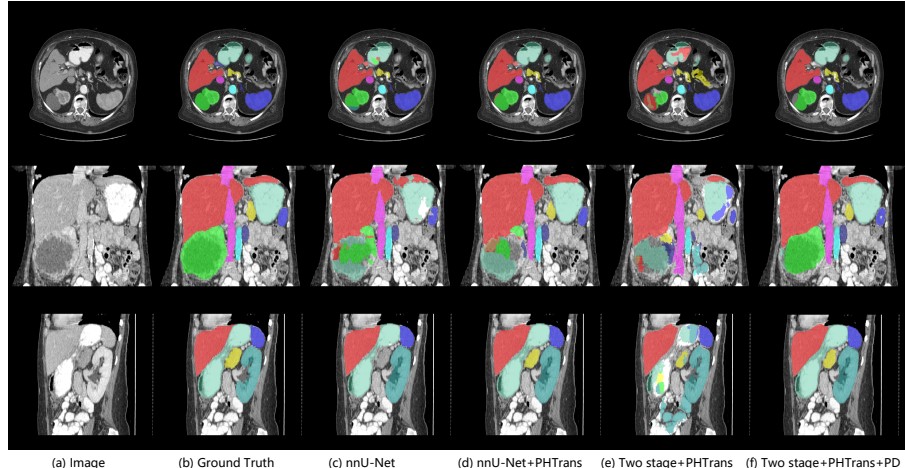

(a) Image    (b) Ground Truth    (c) nnU-Net    (d) nnU-Net+PHTrans    (e) Two stage+PHTrans    (f) Two stage+PHTrans+PD

**Fig. 3.** Visualization of segmentation results of abdominal organs.

### 4.2 Qualitative results on validation set

We visualize the segmentation results of the validation set. The representative samples in Figure 3 demonstrate the success of identifying organ details by TPP, which is the closest to the ground truth compared to other methods due to retaining most of the spatial information of abdominal organs. In particular, it outperforms TP significantly by leveraging unlabeled data, which enhances the robustness of the segmentation model. Furthermore, we show representative examples of poor segmentation, as shown in Figure 4. The first row demonstrates that TP and TPP only detect part of the spleen (blue region), which can be inferred from the flat edge on the right that the first-stage coarse segmentation did not accurately locate the entire abdominal organ. The second row shows a case with a large tumor inside the liver (red region), where the pathological changes pose an extreme challenge for the abdominal organ segmentation. In this case, nnU-Net shows poor segmentation performance. PHTrans improves this result with the global modeling capabilities of the Transformer. In contrast, benefiting from the strategy of whole-volume based input, TP segments a relatively complete liver. However, the performance of TPP to segment livers with pathological changes degrades when training with more pseudo-labeled data. We consider that pseudo labels generated by training on labeled data without liver

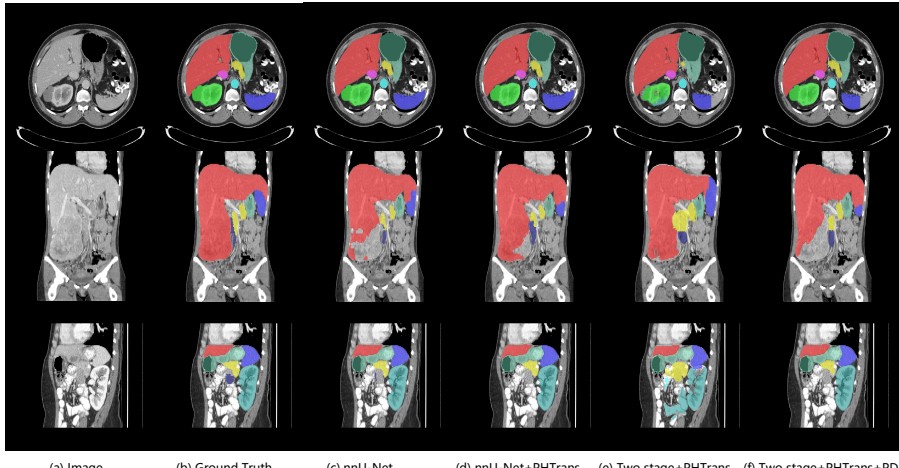

(a) Image     (b) Ground Truth     (c) nnU-Net     (d) nnU-Net+PHTrans     (e) Two stage+PHTrans     (f) Two stage+PHTrans+PD

**Fig. 4.** Challenging examples from validation set.

disease tend to produce a large amount of erroneous label information in liver disease cases, resulting in a degraded performance of TPP for segmenting liver disease cases. The third row shows another case where none of the methods accurately detected the duodenum (blue-black region).

### 4.3 Segmentation efficiency results on validation set

In the official segmentation efficiency evaluation under development environments shown in Table 2, the average inference time of 50 cases in the validation set is 18.62 s, the average maximum GPU memory is 1995.04 MB, and the area under the GPU memory-time curve and the average area under the CPU utilization-time curve are 23196.84 and 319.67, respectively.

### 4.4 Results on final testing set

We submitted the docker of our solution, which was evaluated by the challenge official on the test set, and the results are shown in tables 6 and 7.

### 4.5 Limitation and future work

We used a simple but effective self-training strategy for pseudo-label generation. Theoretically, the accuracy of pseudo labels can be further improved by multiple iterative optimization of the teacher and the student[12] or by performing selective re-training via prioritizing reliable unlabeled datas[17]. In addition, we did not use model deployment in inference. In future work, we plan to use ONNX and TensorRT to further accelerate inference and reduce GPU memory.

**Table 6.** The DSC of the test set from the official evaluation.

| Mean | Liver | RK | Spleen | Pancreas | Aorta | IVC |
|---|---|---|---|---|---|---|
| 0.8941 | 0.9786 | 0.9567 | 0.9377 | 0.828 | 0.9607 | 0.9269 |
| RAG | LAG | Gallbladder | Esophagus | Stomach | Duodenum | LK |
| 0.8749 | 0.861 | 0.8444 | 0.8075 | 0.9281 | 0.7835 | 0.9358 |

**Table 7.** The NSD of the test set from the official evaluation.

| Mean | Liver | RK | Spleen | Pancreas | Aorta | IVC |
|---|---|---|---|---|---|---|
| 0.9395 | 0.9844 | 0.9615 | 0.9416 | 0.9316 | 0.9834 | 0.9395 |
| RAG | LAG | Gallbladder | Esophagus | Stomach | Duodenum | LK |
| 0.9668 | 0.9516 | 0.8556 | 0.8954 | 0.954 | 0.9078 | 0.9409 |

## 5 Conclusion

In this work, we follow the self-training strategy and employ a high-performance PHTrans with the nnU-Net frame for the teacher model to generate precise pseudo-labels. After that, we introduce them with labeled data into a two-stage segmentation framework with lightweight PHTrans for training to improve the performance and generalization ability of the model while remaining efficient. Experiments on the validation set of FLARE2022 demonstrate that our method achieves excellent segmentation performance and computational efficiency. In the future, we will optimize the self-training strategy and apply model deployment to further improve the segmentation performance and fast and low-resource inference.

**Acknowledgements** The authors of this paper declare that the segmentation method they implemented for participation in the FLARE 2022 challenge has not used any pre-trained models nor additional datasets other than those provided by the organizers. The proposed solution is fully automatic without any manual intervention.

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
