# OpenReview forum: "Combining Self-Training and Hybrid Architecture for Semi-supervised Abdominal Organ Segmentation"
_MICCAI.org/2022/Challenge/FLARE_

### Official Review · Reviewer_voFf · 2022-09-14
**The text of this article needs further polishing.**

**Rating:** 9
**Confidence:** 4

**Review:**

Comments to the Author

Based on previous work, the author combines self-training and hybrid architecture for organ segmentation, and achieves good accuracy and inference speed.

- Abstract line 16, "HSD" should be changed to "NSD", the same problem occurs on page 8, section 4.1, the second to last line.
- Page 2, paragraph 2, last seven lines, "have" should be changed to "has"; Change "which" to "that" in the sixth line from the bottom.
- The font in Fig. 1 is too small to read and print easily.
- The 'PD' in Table 3 should be explained, similarly, the abbreviations of organ names in Table 3 and Table 4 should be explained with remarks.
- The text of this article needs further polishing.

Please go through the paper and improve the wording.

---

> ### Author Response · Authors · 2022-10-20
> **We have revised manuscript and provide a point-by-point response to each of the issues raised by reviewer voFf (Reviwer #6)**
>
> We sincerely thank you for the great efforts on our manuscript. According to the valuable suggestions, we have made careful modifications to the manuscript. In what follows, we will provide detailed point-to-point responses to address the reviewers’ concerns.
>
> Comment 1: Abstract line 16, "HSD" should be changed to "NSD", the same problem occurs on page 8, section 4.1, the second to last line.
>
> Response 1:  We are really sorry for our careless mistakes. We have corrected it.
>
> Comment 2: Page 2, paragraph 2, last seven lines, "have" should be changed to "has"; Change "which" to "that" in the sixth line from the bottom.
>
> Response 2:  Thanks for your reminder. We have corrected it.
>
> Comment 3: The font in Fig. 1 is too small to read and print easily.
>
> Response 3: We have enlarged the font in the revised manuscript.
>
> Comment 4: The 'PD' in Table 3 should be explained, similarly, the abbreviations of organ names in Table 3 and Table 4 should be explained with remarks.
>
> Response 4: We have explained it in the header and in the text. (PD: pseudo-labeled data participated in the training）
>
> Comment 5: The text of this article needs further polishing.
>
> Response 5: We have re-edited the manuscript for proper English language, grammar, and punctuation.

---

### Official Review · Reviewer_X1CM · 2022-09-14
**Self-Training and Hybrid Architecture for Semi-supervised Segmentation**

**Rating:** 7
**Confidence:** 4

**Review:**

Summary:

In this paper, the authors propose a new semi-supervised method (PHTrans) to complete the abdominal organ segmentation task through a two-stage segmentation framework.

Strengths:

A hybrid architecture (PHTrans) consisting of CNN and SWN converters is used to generate pseudo labels. All the data is then trained using lightweight PHTrans. The region of interest is located, and both global and local features are taken into account.

Weaknesses:

The ablation test was not comprehensive.
Some parameters in the article are not given a reasonable explanation.

Details:

1. It does not explain how PHTrans-L and PHTrans-S change the calculation amount, and it does not explain the effectiveness of Pure Conv Module.
2. The need for multiple outputs N1+N2 is not explained.
3. The IN operation in the Conv Block in Figure (b) is not explained.
4. In Formula (2), the input of ST module is the ADD operation, and the input of Conv module is the CONCAT operation. No explanation is given.
5. What is the shorthand for PD in Tables 3 and 4 not given.
6. The English of your manuscript must be improved before resubmission. Many sentences contain grammatical and/or spelling mistakes or are not complete sentences.

---

> ### Author Response · Authors · 2022-10-20
> **We have revised manuscript and provide a point-by-point response to each of the issues raised by reviewer voFf (Reviwer #6)**
>
> We sincerely thank you for the great efforts on our manuscript. According to the valuable suggestions, we have made careful modifications to the manuscript. In what follows, we will provide detailed point-to-point responses to address your concerns.
>
> Comment 1: It does not explain how PHTrans-L and PHTrans-S change the calculation amount, and it does not explain the effectiveness of Pure Conv Module.
>
> Comment 2: The need for multiple outputs N1+N2 is not explained.
>
> Comment 3: The IN operation in the Conv Block in Figure (b) is not explained.
>
> Comment 4: In Formula (2), the input of ST module is the ADD operation, and the input of Conv module is the CONCAT operation. No explanation is given.
>
> Response 1,2,3,4: We have added Table 1 (The parameter settings of PHTrans-L and PHTrans-S) to the revised manuscript to illustrate the difference in the parameter settings of PHTrans-L and PHTrans-S. PHTrans is one of our previous works. Details can be found at this [link](https://link.springer.com/chapter/10.1007/978-3-031-16443-9_23).
>
> Comment 5: What is the shorthand for PD in Tables 3 and 4 not given.
>
> Response 5: Thanks for your reminder. We have explained it in the header and in the text. (PD: pseudo-labeled data participated in the training）
>
> Comment 6: The English of your manuscript must be improved before resubmission. Many sentences contain grammatical and/or spelling mistakes or are not complete sentences.
>
> Response 6: We have re-edited the manuscript for proper English language, grammar, and punctuation.

---

### Official Review · Reviewer_rgdU · 2022-09-15
**Impressive work, but still have some thing confusing**

**Rating:** 10
**Confidence:** 3

**Review:**

* In Fig 1, It seem the author didn't talk a lot about generation of pseudo labels. From the figure, I can see that the author using both nn-Unet and PHTrans-L but author didn't explain how this two network get pseudo label.

At last, great work, impressive.

---

> ### Author Response · Authors · 2022-10-20
> **We have revised manuscript and provide a point-by-point response to each of the issues raised by reviewer  rgdU (Reviwer #4)**
>
> We sincerely thank you for the great efforts on our manuscript. According to the valuable suggestions, we have made careful modifications to the manuscript. In what follows, we will provide detailed point-to-point responses to address the reviewers’ concerns.
>
> Comment 1: In Fig 1, It seem the author didn't talk a lot about generation of pseudo labels. From the figure, I can see that the author using both nn-Unet and PHTrans-L but author didn't explain how this two network get pseudo label.
>
> Response 1: In what follows, we have revised the manuscript to address your concerns.
> - "(1) train a teacher model using labeled data, (2) generate pseudo labels for unlabeled data."
> - "We employ one high-performance PHTrans and two lightweight PHTrans as teacher and student models, respectively. (1) and (2) are performed in the nn-UNet framework with the default configuration except that UNet is replaced by PHTrans.

---

### Official Review · Reviewer_oST1 · 2022-09-16
**The authors followed the self-training strategy and employed a high-performance hybrid architecture (PHTrans) consisting of CNN and Swin Transformer for the teacher model to generate precise pseudo labels for unlabeled data.The proposed model achieved average DSC and NSD is 0.8956 and 0.9316, respectively.**

**Rating:** 8
**Confidence:** 5

**Review:**

The authors followed the self-training strategy and employed a high-performance hybrid architecture (PHTrans) consisting of CNN and Swin Transformer for the teacher model to generate precise pseudo labels for unlabeled data. Further, introduced them with labeled data into a two-stage segmentation framework with lightweight PHTrans for training to improve the performance and generalization ability of the model while remaining efficient. The proposed model achieved average DSC and NSD is 0.8956 and 0.9316, respectively. The quality, clarity, and description of the paper are good except for some minor issues.

On the leaderboard, you have a mean DSC of 0.8981, while this manuscript shows a mean DSC of 0.8956. Please address this accordingly.

Tables 3 and 4 are splits that are not recommended. Therefore we suggest constructing a unified table that provides a clearer view to readers.

Make sure you use nnU-Net instead of nnUNet, as in Table 1.

Please provide authentic reference/references for “Connected component-based post-processing”

What do you mean by HSD? Did you mean NSD? If so, please change it to NSD in the abstract and in the text below Table 4. Please check the whole manuscript for English grammar and typos.

---

> ### Author Response · Authors · 2022-10-20
> **We have revised manuscript and provide a point-by-point response to each of the issues raised by reviewer oST1 (Reviwer #3)**
>
> We sincerely thank you for the great efforts on our manuscript. According to the valuable suggestions, we have made careful modifications to the manuscript. In what follows, we will provide detailed point-to-point responses to address the reviewers’ concerns.
>
> Comment 1: On the leaderboard, you have a mean DSC of 0.8981, while this manuscript shows a mean DSC of 0.8956. Please address this accordingly.
>
> Response 1：The mean DSC of 0.8956 is our best result on the leaderboard, which benefits from the large size of the input patch (128, 256, 256). However, to achieve faster inference, we adopt smaller size patches (96, 192, 192).
>
> Comment 2：Tables 3 and 4 are splits that are not recommended. Therefore we suggest constructing a unified table that provides a clearer view to readers.
>
> Response 2：The official daily submission system only has DSC evaluation, not HSD. The evaluation of HSD is from the results of val docker that we only submitted once. So no ablation study was performed on HSD and tables 3 and 4 are split.
>
> Comment 3: Make sure you use nnU-Net instead of nnUNet, as in Table 1.
>
> Response 3: Thank you for your reminder. We have corrected it.
>
> Comment 4: Please provide authentic reference/references for “Connected component-based post-processing”
>
> Response 4: We have added.
>
> Comment 5: What do you mean by HSD? Did you mean NSD? If so, please change it to NSD in the abstract and in the text below Table 4. Please check the whole manuscript for English grammar and typos.
>
> Response 5: We are really sorry for our careless mistakes. We have corrected it.

---

### Official Review · Reviewer_BLJQ · 2022-09-18
**MICCAI-FLARE**

**Rating:** 10
**Confidence:** 4

**Review:**

Advance:
1. The experimental analysis is good, and the analysis of the use of unlabeled and labeled data is sufficiently credible.
2. The network architecture is based on CNN and transformer is very creative and has strong performance.

---

### Official Review · Reviewer_WxwE · 2022-09-21
**Combining Self-Training and Hybrid Architecture for Semi-supervised Abdominal Organ Segmentation**

**Rating:** 10
**Confidence:** 5

**Review:**

Pros:
- self-training strategy, a high-performance hybrid architecture consisting of CNN and Swin Transformer
- high dice
- overall, authors did a great job

---

### Meta-Review · Program_Chairs · 2022-09-28

**Recommendation:** Minor Revision
**Confidence:** 5

**Metareview:**

Nice paper. Please address the reviewers' comments in the revised manuscript.